# Crack Detection of Bridge Concrete Components Based on Large-Scene Images Using an Unmanned Aerial Vehicle

**DOI:** 10.3390/s23146271

**Published:** 2023-07-10

**Authors:** Zhen Xu, Yingwang Wang, Xintian Hao, Jingjing Fan

**Affiliations:** 1School of Civil and Resource Engineering, University of Science and Technology Beijing, Beijing 100083, China; m202110055@xs.ustb.edu.cn (Y.W.); s20200036@xs.ustb.edu.cn (X.H.); m202220122@xs.ustb.edu.cn (J.F.); 2Beijing Advanced Innovation Center for Materials Genome Engineering, University of Science and Technology Beijing, Beijing 100083, China

**Keywords:** UAV, large-scene images, crack detection, high noise, width calculation

## Abstract

The current method of crack detection in bridges using unmanned aerial vehicles (UAVs) relies heavily on acquiring local images of bridge concrete components, making image acquisition inefficient. To address this, we propose a crack detection method that utilizes large-scene images acquired by a UAV. First, our approach involves designing a UAV-based scheme for acquiring large-scene images of bridges, followed by processing these images using a background denoising algorithm. Subsequently, we use a maximum crack width calculation algorithm that is based on the region of interest and the maximum inscribed circle. Finally, we applied the method to a typical reinforced concrete bridge. The results show that the large-scene images are only 1/9–1/22 of the local images for this bridge, which significantly improves detection efficiency. Moreover, the accuracy of the crack detection can reach up to 93.4%.

## 1. Introduction

Concrete structural bridges are abundant. The investigation of structural safety and durability in existing reinforced concrete bridges stands as a paramount research area within the field of structural engineering [1,2]. However, cracks usually form during their full-life cycle. These cracks can reduce the durability of bridges and even affect their structural safety [3]. Consequently, the research on crack detection for concrete structural bridges is of great significance.

With the development of unmanned aerial vehicle (UAV) technology, an increasing number of researchers have employed UAV images to facilitate bridge crack detection. Relevant research involves various bridge crack detection technologies, such as visual inspection [4,5], deep learning-based recognition [6,7], the crack central point method [8], and attention-based crack detection [9]. Currently, in UAV-based crack detection, image acquisition predominantly relies on localized component photographs. Numerous scholars have utilized UAVs for close-range photography, capturing high-definition images of local concrete components of bridges [10,11,12,13,14,15]. While the mean average precision (mAP) in crack detection of these localized images is considerably high, the dimensions of the bridge components corresponding to each image are relatively small. For large-scale bridges, a large number of local images need to be collected, resulting in low detection efficiency. Therefore, to improve the efficiency of image acquisition, UAV-based large-scene images can be employed for crack detection, to take fewer photos of the entire bridge.

However, large-scene images have the problem of high noise caused by complex backgrounds [16]. In previous image-based crack detection research, the majority employed noise-free or low-noise concrete component images captured at a close range [17,18]. However, in large-scene images acquired via UAV, the actual bridge environment is complex, so the surrounding trees, buildings, and pipelines may interfere with crack detection. Therefore, unlike the low-noise or noise-free localized concrete bridge component images, large-scene images contain lots of background noise, and it is necessary to solve the problem of crack detection caused by high noise.

Moreover, within large-scene images, the pixels corresponding to components are relatively low, which may lead to a decrease in crack detection accuracy [19]. However, the safety thresholds of crack width vary across different bridge components. For components with larger crack width thresholds, UAV-based large-scene images can be used. According to China’s “Technical Standard of Maintenance for City Bridge” [20] and “Technical Code for Test and Evaluation of City Bridges” [21], the crack width threshold for bridge pavements, reinforced concrete beams, piers, and crash barriers are 3 mm, 0.3 mm, 0.4 mm, and 5 mm, respectively. Consequently, for bridge pavements and crash barriers, large-scene images can be employed for crack detection. However, low pixels still exacerbate the difficulty of crack detection and width calculation [22], which requires specific research.

To address these concerns, this study proposes a method for detecting cracks in concrete bridge components using UAV-based large-scene images. First, a UAV-based acquisition scheme of large-scene images for bridges and a background denoising algorithm for large-scene images are designed so that the cracks in large-scene images are accurately detected. Subsequently, an algorithm for calculating the maximum crack width is designed based on the region of interest and the maximum inscribed circle. Finally, the effectiveness of the proposed method is validated through the detection of a typical reinforced concrete beam bridge, achieving a high detection efficiency based on UAV-based large-scene images and offering a novel solution for the inspection of bridge components with larger thresholds of crack width.

## 2. Technical Framework

The technical framework of this study is illustrated in Figure 1, which consists of three parts:(1)Crack detection. Firstly, UAVs are used to acquire large-scene images of bridge structures containing multiple components. Subsequently, grid segmentation and classification networks are combined to denoise large-scale environmental backgrounds, thereby enhancing the accuracy of crack detection. Lastly, the YOLOv5 algorithm is adopted for noise-resistant crack detection in bridges.(2)Crack width calculation. Firstly, the region of interest (ROI) of cracks is cropped according to the crack detection results, reducing the research area and mitigating the interference of background noise on crack width calculation. The ROI image is then enhanced, augmenting the image resolution. Finally, the maximum crack width is determined based on the principle of a maximum inscribed circle within the contour, and precision control conditions satisfying the crack width thresholds are established through experiments.(3)Case study. A reinforced concrete girder bridge is selected as a case study. The cracks of crash barriers are detected and their widths are also calculated, demonstrating the advantages of the proposed method. This study mainly focuses on bridge components with a large crack width threshold, especially for bridge pavements and crash barriers, in which crack width thresholds for bridge pavements and crash barriers are 3 mm and 5 mm, respectively. Based on the threshold values of crack widths for different bridge components, our method provides precision control conditions with a crack width calculation error of 5% in Section 3.2. In addition, in non-contact crack detection in civil engineering, it is generally desired to achieve an mAP of 90% or higher [23,24,25]. This implies the aspiration to accurately discern between crack and non-crack regions.

## 3. Crack Detection Based on Large-Scene Images

### 3.1. Acquisition of UAV-Based Large-Scene Images

This study faces several challenges in the acquisition of large-scene images. For instance, during the collection process, the presence of occlusions in the field of view, obstacles affecting UAV flights, intense or low-lighting conditions, and other adverse environmental factors can severely impact the quality of the captured photographs. Consequently, these factors can significantly reduce the efficiency of image acquisition and compromise the accuracy of crack detection.

The main purpose of employing UAVs for bridge large-scene image acquisition is to devise aerial photography routes. As most bridges exhibit a larger span with length surpassing width, they are approximately rectangular. As illustrated in Figure 2, an aerial photography route planning is executed according to the flight overlap rate, ensuring the comprehensive acquisition of bridge components in the length direction of the bridge.

The maximum spacing distance between two adjacent aerial photography points (i.e., *d*) can be calculated, taking into account camera parameters, vertical photography distance, and image overlap. As shown in Figure 3, points *A* and *B* represent two adjacent aerial photography locations, with *d* as the aerial photography spacing, *f* as the camera focal length, *l* as the image sensor size in the flight direction, *D* as the vertical distance from the camera to the surface of the photographed object, *L* as the actual physical length of the aerial photography image, and $L^{\prime }$ as the overlap length.

When employing a fixed focal length lens UAV, both *f* and *l* remain constant. Based on the proportional relationship shown in Figure 3, the relationship between *L* and *D* is as follows (1):(1)$$\frac{l}{f}=\frac{L}{D}$$

During the aerial photography process, the image overlap rate *R* can be determined using Equation (2):(2)$$R=\frac{L^{\prime }}{L}=\frac{L-d}{L}=1-\frac{d}{L}$$

Based on actual shooting tests, it is recommended to maintain an *R* between 5 and 10%. The aerial photography spacing *d* can be calculated using Equation (3):(3)$$d=\left(1-R\right)L=\frac{lD}{f}\left(1-R\right)$$

The camera’s vertical distance *D* can be ascertained according to the precision control conditions of crack detections, as detailed in Section 4.3 of this study.

### 3.2. Background Denoising Based on Grid Segmentation and Classification Network

Classification networks typically employ small-sized images for object classification. Grid segmentation, on the other hand, can segment large-scene images into small-sized images, effectively eliminating pure background images.

To tackle the challenge of crack detection caused by complex environmental backgrounds in large-scale images, this study proposes a solution that combines grid segmentation and a classification network, thereby achieving extensive background denoising. The specific workflow is illustrated in Figure 4.

Firstly, large-scene images are segmented into equally sized sub-images via a grid, which serves as the dataset for background denoising and crack detection in the classification network. An excessively large grid size may prevent the effective separation of the background and target bridge structures, resulting in inadequate background denoising. Conversely, if the grid size is too small, the images will be overly fragmented, making it difficult to reflect the characteristics of the bridge and may increase the computational load for classification. Given that the prevalent image size for existing crack detection algorithms ranges from 200 to 320 pixels, this study recommends partitioning large-scene images into equally sized sub-images within this range.

Subsequently, the classification network segregates sub-images into pure background images and those containing concrete components. The background is denoised by eliminating the pure background sub-images, as shown in Figure 5. Conventional classification networks (e.g., convolutional neural networks, CNNs) require substantial training samples and slow training efficiency for complex classification problems. To balance computational efficiency and accuracy, this study employs MobileNets for classification. As a lightweight deep neural network, MobileNets significantly reduces model parameters and computational load compared to CNNs, with only a slight decrease in accuracy [26].

In this study, a classification dataset comprising 1026 images containing concrete components and 944 pure background images is assembled. The dataset is divided into training and validation sets at an 8:2 ratio. The training epochs are set to 300, and MobileNets separable convolution is used to accelerate the training process, resulting in a classification model. Upon validation, both precision and recall of the classification reach 98.85%, indicating the model’s exceptional classification performance.

### 3.3. Noise-Resistant Crack Detection Based on YOLOv5

Current crack detection methods primarily employ sample sets of noise-free or low-noise images. However, these methods are unsuitable for the crack detection of large-scene images with high noise. Consequently, it is necessary to construct a crack detection dataset suitable for large-scene, high-noise images of bridges.

#### 3.3.1. High-Noise Dataset Construction

This study employs the open-source image annotation software LabelMe to delineate cracks using bounding boxes, and annotations are added to all cracks in the bridge sub-images. In large-scene images acquired by drone aerial photography, cracks are extremely subtle. Therefore, to ensure accurate detection, the smallest bounding box that entirely encompasses a single crack should be used during the annotation process. Based on the large-scene images acquired by UAV, 2906 concrete images containing cracks obtained after background denoising through classification are selected to construct the high-noise crack dataset and they are annotated by employing LabelMe.

To enhance model training efficacy and avoid overfitting due to data scarcity, the annotated samples are expanded to 5910 through a data augmentation approach, which constitutes a crack detection sample set suitable for large-scale high-noise images of bridges.

#### 3.3.2. Model Training of Crack Detection

YOLO (you only look once) employs a single network to directly detect targets and generate location coordinates, thereby ensuring exceptional detection efficiency [27]. In this study, the YOLOv5 algorithm is employed for crack detection.

This study adopts transfer learning to train the YOLOv5-based crack detection model. Specifically, the dataset is divided into training and validation subsets at an 8:2 ratio, and the optimal bounding box values within the training subset are adaptively calculated. Parameter fine-tuning is performed based on the pre-trained model, and the network weights recommended by the YOLOv5 officials are selected. The input volume for each batch is determined according to the performance of the hardware device, and the training subset is input into the network for iteratively training the bridge crack detection model.

mAP is a widely used metric for comprehensively evaluating the performance of object detection models [28]. The AP curve typically exhibits fluctuations, but by interpolating all points (e.g., Equations (4) and (5)), a smoother curve is obtained. The mAP can be seen as an approximation of the area under the precision–recall (PR) curve (as in Equation (6)), aiming to mitigate the impact of curve fluctuations.

For each class, the AP is calculated, and then the mAP is obtained by averaging across all classes using Equations (4) and (5):(4)$$\mathrm{AP}=\frac{\sum_{k=1}^{n}P(k)\times rel(k)}{M}$$
(5)$$\mathrm{mAP}=\sum_{i}^{Q}\frac{{\mathrm{AP}}_{i}}{Q}$$

In this context, *P*(*k*) represents the precision of the first *k* results, while *rel*(*k*) indicates whether the *k*^th^ result belongs to the relevant category (1 for relevant, 0 for irrelevant). *M* represents the total number of instances in that category, and *Q* represents the total number of categories.

The equation for calculating the area under the PR curve is Equation (6):(6)$$\sum_{n}(R_{n}-R_{n-1})P_{n}$$

In this context, *R_n_* denotes the recall rate associated with the *n*^th^ data point, while *P_n_* represents the precision associated with the *n*^th^ data point.

The PR curve is calculated based on the training results, as shown in Figure 6. The mAP value of the crack detection model in this study reaches an impressive 93.4%, indicating the high accuracy of the model.

## 4. Crack Width Calculation Based on ROI and Maximum Inscribed Circle

In large-scene images, the pixels corresponding to structural components are relatively low, and factors such as surface shadows and contaminants may interfere with the calculation of image width. This study proposes an algorithm that involves cropping the crack ROI to minimize the research scope, thereby reducing the impact of factors such as shadows and contamination on crack width calculation. Subsequently, pixel enhancement is employed to enhance the pixels of the ROI images, and the maximum inscribed circle method of the contour is adopted to calculate the maximum width of the crack.

### 4.1. Cropping of ROI in Crack Image 

The crack detection bounding box, obtained from crack detection, serves as the ROI. The position information file of the detection bounding box is traversed, and the content of each file is automatically stored as an array. These arrays are assigned in a list. The specific crack information in the list includes the length, width, and height of the detected bounding box and the coordinates of the center of the box. The crack detection bounding box information is then input into the cropping algorithm to obtain the ROI with cracks. The entire procedure is shown in Figure 7.

### 4.2. Crack Width Calculation Based on Maximum Inscribed Circle

In large-scene images, the pixels corresponding to structural components are relatively low, potentially resulting in unclear crack edges. The pixel enhancement can achieve high-resolution processing of crack images, offering two primary advantages. First, it sharpens the crack edges and enhances the contrast between the cracks and the concrete background, and better determines the crack contour. Second, it increases the pixel resolution, addressing the issue of inaccurate crack width calculations due to indivisible pixels.

Based on the high-resolution ROI images, this study continuously updates the maximum width value by cyclically searching the inscribed circles of the crack contour, and finally shows the maximum crack width and its corresponding circle center coordinates, as shown in Figure 8. First, the cropped ROI image is binarized and the crack contour is identified. Then, 1/10–1/5 of internal pixel points of the contour are randomly selected, calculating and updating the maximum inscribed circle of the contour. Finally, the maximum crack width is outputted and visual annotation is performed, as shown in Figure 9.

The crack width calculation mentioned above is in pixels. The actual physical width of the crack $W_{a}$ is computed according to Equation (7):(7)$$\frac{W_{a}}{W_{p}}=\frac{L}{W_{i}}$$
where $W_{p}$ denotes the pixel width of the crack, $W_{i}$ represents the pixel width of the large-scene image, and *L* signifies the actual physical length corresponding to the large-scene image in Figure 3.

### 4.3. Precision Control

To meet the requirements of the crack width threshold to be inspected, it is crucial to regulate the vertical photography distance of the UAV. An excessive vertical photography distance will result in low pixels of the crack image, and thus possibly cause a considerable error in the crack width calculation.

This study conducts a series of tests on the accuracy of crack width and the vertical photography distance. The crack width thresholds of 3 mm and 5 mm, corresponding to the bridge pavements and crash barrier, are employed as the tests. A DJI Phantom 4 UAV, equipped with a common camera (see Table 1), is used for capturing images. Furthermore, to compare the precision in detecting the crack between an industrial camera and a common camera, a DJI Mavic 2 Enterprise Advanced UAV equipped with industrial cameras was added for a comparative experiment. Considering the visual obstacle avoidance distance requirements for this UAV during flight, the two cracks with widths of 3 mm and 5 mm were photographed at vertical photography distances of 2 m, 2.5 m, 3 m, 3.5 m, 4 m, 4.5 m, and 5 m, respectively.

For each vertical photography distance, five images were acquired for the two cracks. Subsequently, the crack ROI area was cropped, and the ROI image resolution was increased using pixel enhancement. This process minimized the error caused by the low pixel count of the large-scene image. The crack width was calculated using the method proposed in this study and compared with the actual width. The average relative error of the five images serves as the crack error. The minimum errors of the crack at different vertical photography distances are obtained, as shown in Table 2.

Table 2 presents the error results obtained from analyzing images acquired by a DJI Phantom 4 UAV, with image pixels of 4000 × 3000. Similarly, Table 3 shows the error results obtained from analyzing images acquired by a DJI Mavic 2 Enterprise Advanced UAV, with image pixels of 8000 × 6000. From both tables, it can be inferred that using a camera with a higher image resolution is advantageous for obtaining reliable crack width calculation results at longer vertical photography distances. Considering the practical engineering applications, the width of cracks determines subsequent maintenance decisions and significantly impacts the structural safety and durability of bridges. Therefore, strict control over the average relative error is necessary, aiming for a value of 5% or lower. As the vertical photography distance increases, reaching the 5% threshold for calculation errors in cracks with smaller widths becomes easier.

According to Table 2, when maintaining crack error control within 5%, the maximum vertical photography distances are 2.5 m for 3 mm cracks and 4.5 m for 5 mm cracks. Similarly, as indicated in Table 3, the maximum vertical photography distances for maintaining crack error control within 5% are determined to be 4.0 m for 3 mm cracks and 5.0 m for 5 mm cracks. Shooting within the maximum vertical photography distance can ensure the accuracy of crack detection.

## 5. Case Study

### 5.1. Case Introduction

This study selects a beam bridge located in Wuqiang County, Hebei Province, which spans Tianpinggou on Ping’an North Road, as a case study. The bridge has an approximate length of 60 m and a width of 16 m, employing a reinforced concrete structure. The proposed method is applied to detect cracks in crash barriers with cracks.

The performance indexes of a DJI Phantom 4 UAV are shown in Table 1. A DJI Phantom 4 UAV, equipped with a Sony Exmor R CMOS image sensor, a camera focal length *f* of 3.43 mm, and an image sensor flight size *l* of 6.17 mm, is utilized for data collection. The acquired images had a resolution of 4000 × 3000 pixels. During the shooting process, an overlap rate of 10% was maintained, with a flight speed of 2 m/s and a flight altitude of 3 m. With a vertical distance *D* of 3 m, the aerial photography spacing *d* was calculated to be 4.86 m. In total, 13 large-scene images were captured on one side.

### 5.2. Background Denoising

Following the proposed method, the collected 13 large-scene images (4000 × 3000 pixels) were batch segmented into 2496 small-sized sub-images (250 × 250 pixels), as depicted in Figure 10. These sub-images are then input into the pre-trained MobileNets to remove background category images. The background denoising result is shown in Figure 10b, where among the 2496 sub-images, a total of 1142 concrete images were obtained.

### 5.3. Crack Detection

The pre-trained YOLOv5 network was employed to detect cracks using the 1142 classified concrete images. As a result, 113 sub-images containing cracks were identified, as shown in Figure 11.

### 5.4. Calculation on Maximum Crack Width 

The method described in Section 4.1 was used to crop the crack image and obtain the crack ROI, as shown in Figure 8. According to the accuracy control requirements outlined in Table 1, the ROI image resolution was first increased fourfold. Subsequently, the maximum inscribed circle of the contour was calculated, and the maximum crack width, along with the corresponding circle center coordinates, was outputted. Finally, the annotation of the maximum crack width was completed, as shown in Figure 12. In total, 28 images with crack widths of 5 mm or more were identified, and these cracks will be manually inspected to ensure the safety of the bridge.

### 5.5. Efficiency and Accuracy Comparison

A comparison of the image acquisition efficiency and crack detection accuracy between our method and current mainstream methods is shown in Table 4. The image acquisition efficiency of the proposed method is significantly higher than that of other methods, with aerial photography efficiency being 9–22 times greater than that of the literature in Table 4. This improvement substantially reduces the number of images required. Furthermore, the mAP in crack detection of the proposed method reaches an impressive 93.4%. Although the method [29] achieves slightly higher accuracy compared to the method used in this study, it is important to note that this is primarily due to the shorter vertical photography distance of 1 m while the orthographic distance of 3.0 m is adopted in ours, so that the image resolution is higher than ours. As a result, the recognition accuracy is improved. However, it is worth mentioning that the overall efficiency of the method is not high. Consequently, for components with larger crack width thresholds, such as bridge pavements and crash barriers, the proposed method offers a clear advantage in image acquisition efficiency while maintaining high detection accuracy. This approach can better serve the operation and maintenance of bridges.

## 6. Conclusions

In this study, we proposed a method for detecting cracks in bridge concrete components based on UAV-based large-scene images and applied it to real bridges. The following conclusions were drawn:(1)For crack detection, the mAP of our method reached up to 93.4% for large-scene UAV images based on the training set made of the denoising images and YOLOv5, which means a high detection accuracy. This is due to the designed background denoising algorithm that combines grid segmentation and a classification network, effectively eliminating the large-scene environmental background.(2)In the aspect of crack width calculation, the errors of crack width can be within 5% in this study when the maximum vertical photography distances are 2.5 m and 4.5 m which satisfies the engineering precision requirements for crack detection, especially for bridge pavements and crash barriers whose crack detection thresholds are 3 mm and 5 mm, respectively.(3)In terms of efficiency, the case study indicates that the image acquisition efficiency of large-scene images is 9–22 times higher than that of existing methods of local images, which is significant for improving the efficiency of crack detection.

Regrettably, our method may have certain limitations in detecting cracks of the bridge components (e.g., reinforced concrete beams, piers) with smaller crack width thresholds. For concrete bridges, different structural elements have varying permissible crack width thresholds. Especially, components with smaller threshold values often necessitate the use of higher-resolution camera equipment or shorter shooting distances, which may potentially result in reduced data acquisition efficiency. However, our method has a considerable application prospect for bridge components with large thresholds of crack detection because it improves image acquisition efficiency while maintaining high detection accuracy of 93.4% and low calculation error within 5%.

## Figures and Tables

**Figure 1 sensors-23-06271-f001:**
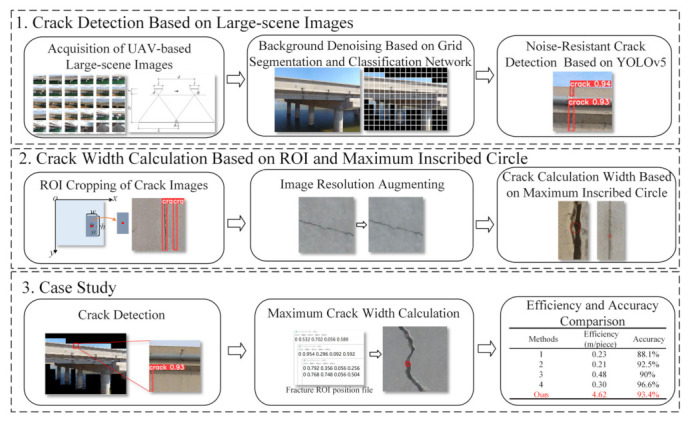
Technical framework.

**Figure 2 sensors-23-06271-f002:**
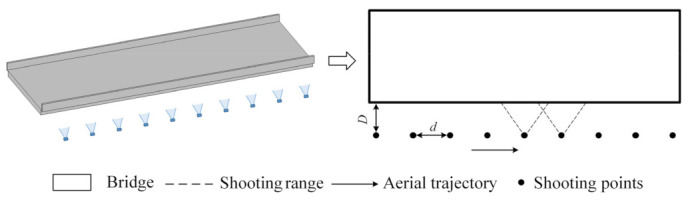
Large-scene image acquisition strategy using UAVs.

**Figure 3 sensors-23-06271-f003:**
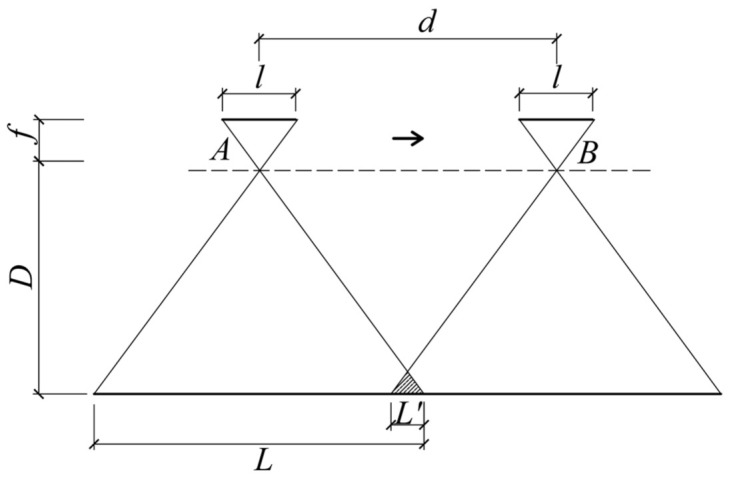
Mathematic model of aerial photography.

**Figure 4 sensors-23-06271-f004:**
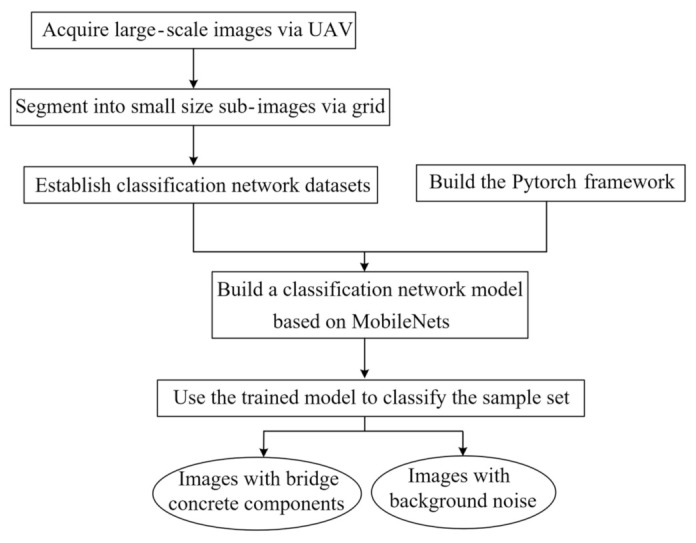
Workflow for background denoising.

**Figure 5 sensors-23-06271-f005:**
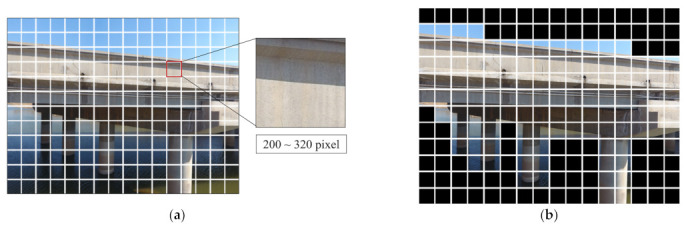
Background denoising of large-scene images of concrete bridges. (**a**) Grid segmentation of large-scene images. (**b**) Part of a concrete bridge member after background denoising.

**Figure 6 sensors-23-06271-f006:**
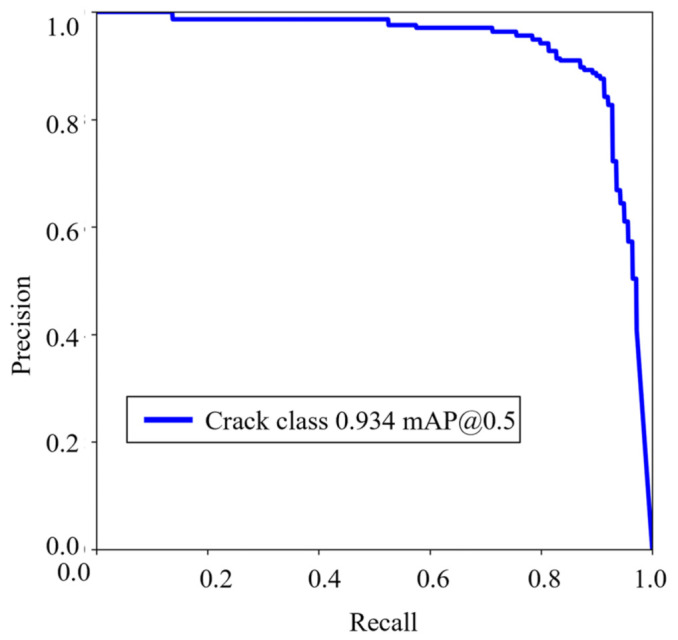
The PR curve of crack detection model.

**Figure 7 sensors-23-06271-f007:**
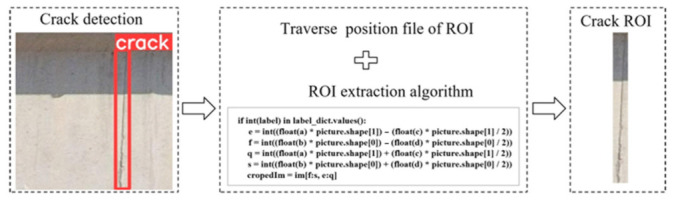
The process of ROI cropping.

**Figure 8 sensors-23-06271-f008:**
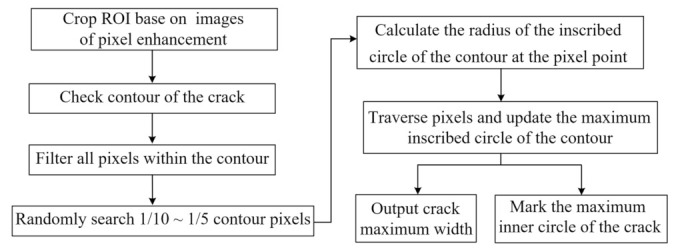
Flowchart for calculating maximum crack width.

**Figure 9 sensors-23-06271-f009:**
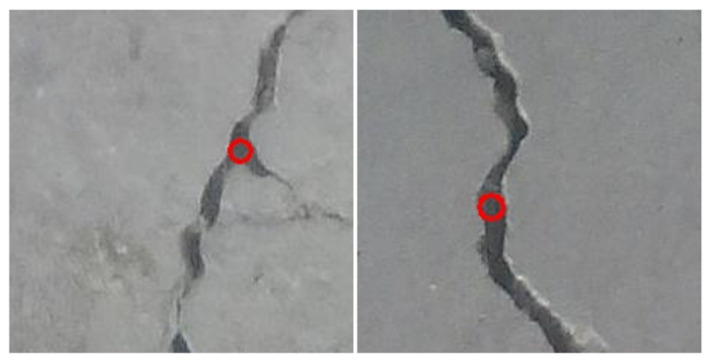
Visual annotation of maximum crack width.

**Figure 10 sensors-23-06271-f010:**
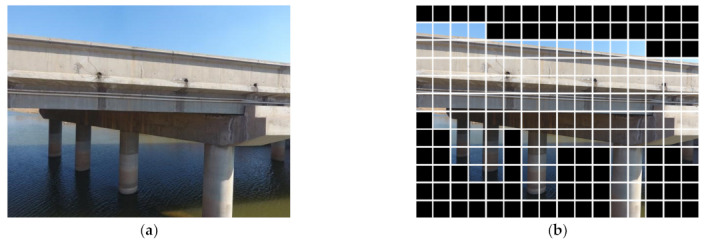
UAV-based large-scene images (**a**) Original images. (**b**) Background denoising results.

**Figure 11 sensors-23-06271-f011:**
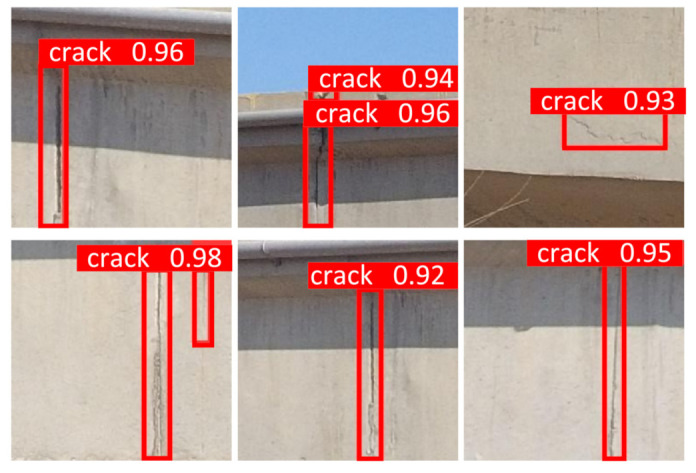
Crack detection results.

**Figure 12 sensors-23-06271-f012:**
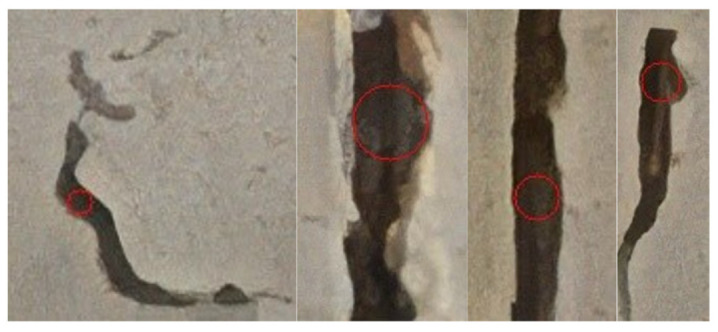
Visual annotation of maximum crack width in small-size sub-images.

**Table 1 sensors-23-06271-t001:** The performance indexes of two UAVs.

Indexes	DJI Phantom 4 UAV	DJI Mavic 2 Enterprise Advanced UAV
Image sensor size	6.17 mm × 4.60 mm	6.4 mm × 3.6 mm
Camera focal length	3.43 mm	4.80 mm
Image pixels	4000 × 3000	8000 × 6000

**Table 2 sensors-23-06271-t002:** Relation between vertical photography distance, resolution magnification, and the minimum error of crack width via a DJI Phantom 4 UAV.

Vertical Photography Distance/m	Crack Width at 3 mm		Crack Width at 5 mm	
	Resolution Magnification	Minimum Error	Resolution Magnification	Minimum Error
2.0	2	4.09%	3	1.38%
2.5	3	0.79%	2	0.40%
3.0	1	11.05%	4	0.81%
3.5	1	4.09%	4	1.99%
4.0	2	19.44%	4	3.74%
4.5	2	30.66%	2	0.443%
5.0	1	26.10%	1	10.33%

**Table 3 sensors-23-06271-t003:** Relation between vertical photography distance, resolution magnification, and the minimum error of crack width via a DJI Mavic 2 Enterprise Advanced UAV.

Vertical Photography Distance/m	Crack width at 3 mm		Crack Width at 5 mm	
	Resolution Magnification	Minimum Error	Resolution Magnification	Minimum Error
2.0	1	0.76%	1	0.50%
2.5	3	0.73%	5	0.99%
3.0	1	0.79%	4	1.64%
3.5	1	3.75%	5	1.00%
4.0	1	9.41%	5	1.70%
4.5	1	0.45%	5	4.37%
5.0	1	10.23%	1	0.67%

**Table 4 sensors-23-06271-t004:** Comparison of image acquisition efficiency and crack detection accuracy.

Methods	Bridge Length (m)	Number of Images (Piece)	Aerial Photography Efficiency (m/Piece)	Detection Accuracy
Method [17]	490	2097	0.23	88.1%
Method [30]	136.8	637	0.21	92.5%
Method [7]	1700	3548	0.48	90%
Method [29]	200	676	0.30	96.6%
Our method	60	13	4.62	93.4%

## Data Availability

Not applicable.

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
