# Peer review of "Crack Detection of Bridge Concrete Components Based on Large-Scene Images Using an Unmanned Aerial Vehicle"

_sensors, 2023, doi:10.3390/s23146271_

Round 1
Reviewer 1 Report
The manuscript titled "Crack Detection of Bridge Concrete Components Based on the Large-Scene Images Using an Unmanned Aerial Vehicle" presents a novel method for crack detection in bridge concrete components using UAV-based large-scene images. The abstract, technical framework, and application case provide a comprehensive overview of the study. However, there are several major comments that should be addressed by the authors before a final decision can be made:
1. The abstract should be more concise and provide a clear summary of the paper's objectives, methods, and results. Currently, it includes repetitive information and can be condensed.
2. It would be helpful to provide a brief explanation of the grid segmentation and classification networks used in the background denoising algorithm. This would enhance the readers' understanding of the methodology.
3. Figure 1, illustrating the technical framework, should be more detailed to clearly depict the different steps involved in crack detection and width calculation. Additional labels and descriptions would be beneficial.
4. In the application case, it is mentioned that a DJI Phantom 4 drone with specific equipment was used for data collection. However, details about the flight parameters (e.g., altitude, speed) and the camera settings (e.g., image resolution) are missing. Including this information would improve reproducibility.
5. The conclusion section could be more structured and highlight the key findings of the study. It should also address the limitations and potential future directions for research.
6. The manuscript should undergo thorough proofreading to correct grammatical errors and improve the overall clarity of the language used.
Once these comments have been addressed, the paper has the potential to make a valuable contribution to the field of crack detection in bridge concrete components using UAV-based large-scene images.
The manuscript should undergo thorough proofreading to correct grammatical errors and improve the overall clarity of the language used.
Author Response
Thanks to the reviewer for your valuable comments. We have made thorough revisions based on your requirements. If you have any questions or suggestions, please feel free to contact us promptly. Thank you once again for your kind work.

Reviewer 2 Report
The article, "Crack Detection of Bridge Concrete Components Based on the Large-Scene Images Using an Unmanned Aerial Vehicle" by Xu Zhen et al., presents an innovative approach to crack detection using UAV-acquired large-scene images, which has the potential to improve efficiency and accuracy. The utilization of UAVs and large-scene images is a promising and novel solution for this task. The conclusion highlights the high detection accuracy achieved, reaching up to 93.4% for large-scene UAV images, demonstrating the effectiveness of the proposed method.
Overall, the article is well-written and presents an innovative approach to crack detection. By incorporating the suggested enhancements, the article can further strengthen its contribution to the field and provide more comprehensive insights for readers.
1. The article lacks specific information about the evaluation metrics used, comparisons with existing methods, and potential limitations of the proposed approach. Providing more comprehensive evaluation details would strengthen the credibility and usefulness of the findings.
2. It would be beneficial to mention any potential limitations or considerations when applying the method to diverse structural contexts. Addressing these aspects would help readers understand the applicability and potential challenges in different bridge scenarios.
3. The article does not mention any challenges or obstacles faced during the implementation or evaluation of the proposed method. Acknowledging and addressing potential challenges would have provided a more comprehensive view of the study's limitations and future directions.
Author Response

(The authors gave the same response as above.)

Reviewer 3 Report
1.Specification of camera has not been described.
2.Can author explain why method [25] in Table 2 exhibit better accuracy than their own method?
3.The reviewer might expect the thorough discussion of precision/percentage error over resolution of image. Industrial camera and common camera can be compared for the precision in detecting the crack.
4.The technical framework and calculation were clearly discussed. The error analysis was discussed too.
5.“For bridge components with large thresholds of crack detection, the proposed method has a considerable application prospect, because it improves image acquisition efficiency while maintaining high detection accuracy and precision.”
The quoted statement above was stated in conclusion. Please specify the improved accuracy and precision after improvement.
Author Response
Thanks to the reviewer for your valuable comments. We have made thorough revisions based on your requirements. If you have any questions or suggestions, please feel free to contact us promptly. Thank you once again for your kind work!

Reviewer 4 Report
The paper “Crack Detection of Bridge Concrete Components Based on the Large-Scene Images Using an Unmanned Aerial Vehicle” reports a research work about the crack detection of bridge based on unmanned aerial vehicles (UAVs). In particular, the authors proposed a new approach based on the large-scene images using an UAV in order to improve the efficiency of images acquisition. The method is applied to a typical reinforced concrete bridge so as to show the efficiency of the approach. In general, the manuscript appears well-organized in its different Sections and the results are clearly commented in the text. For these reasons, it is opinion of this reviewer that the paper can be considered for the publication in Sensors journal after the following minor improvements/corrections:
1) lines 25-27, consider as reported in 10.1016/j.engfailanal.2022.106546 and 10.1016/j.engfailanal.2020.104727 for the structural safety of existing reinforced concrete bridges subject to horizontal actions;
2) Section 2: specify precision and tolerance of the crack detection (depth, dimensions, etc..);
3) Figure 6: Are the two lines overlapping?;
4) Improve the quality of the Figure 7;
5) Section 4.3: add a new Table with the characteristics of the instrumentation used;
6) lines 263, 264: The sentence: “During the shooting process, the overlap rate is maintained at 10%..” is repeated;
7) line 265: What do the authors mean by the sentence “Based on the proposed method”?;
8) Improve the conclusions adding the limits of the proposed approach and highlighting the original aspects.
A moderate improvements of English is suggested.
Author Response
Thanks to the reviewer for your valuable comments. We have made thorough revisions based on your requirements. If you have any questions or suggestions, please feel free to contact us promptly.

Round 2
Reviewer 1 Report
The paper can now be accepted for publication.